# Be Aware of Burnout! The Role of Changes in Academic Burnout in Problematic Facebook Usage among University Students

**DOI:** 10.3390/ijerph18158055

**Published:** 2021-07-29

**Authors:** Katarzyna Tomaszek, Agnieszka Muchacka-Cymerman

**Affiliations:** Department of Pedagogy and Psychology, Pedagogical University of Kraków, 30-084 Kraków, Poland; katarzyna.tomaszek@up.krakow.pl

**Keywords:** Facebook intrusion, academic burnout, Facebook motives and importance, interaction effects

## Abstract

Most previous research has examined the relationship between FB addiction and burnout level by conducting cross-sectional studies. Little is known about the impact of changes in burnout on FB addiction in an educational context. Through a two-way longitudinal survey of a student population sample (*N* = 115), this study examined the influence of changes in academic burnout over time and FB motives and importance (measured at the beginning and the end of the semester) on FB intrusion measured at the end of the academic semester. The findings show that: (1) increases in cynicism and in FB motives and importance significantly predicted time2 FB intrusion; (2) FB importance enhanced the prediction power of changes in the academic burnout total score, exhaustion and personal inefficacy, and reduced the regression coefficient of changes in cynicism; (3) the interaction effects between FB social motive use and changes in academic burnout, as well as between FB importance and personal inefficacy and exhaustion, accounted for a significant change in the explained variance of time2 FB intrusion. About 20–30% of the variance in time2 FB intrusion was explained by all the examined variables and by the interactions between them. The results suggest that changes in academic burnout and FB motives and importance are suppressive variables, as including these variables in the regression model all together changed the significance of the relationship between independent variables and FB intrusion.

## 1. Introduction

An increasing number of studies have been carried out to investigate the antecedents and consequences of social media addiction among university students. The study conducted by Khan [1] suggested that even more than half of university students present mild to heavy symptoms of FB addiction. The syndrome of academic burnout is also a widespread condition among students. For example, according to a meta-analysis conducted by Rosales–Ricardo et al. [2], the prevalence of each dimension of academic burnout syndrome can be estimated at 55.4% for emotional exhaustion, 31.6% for cynicism and 30.9% for academic efficacy. Its symptoms are analogous to those of workers, i.e., higher absenteeism, higher dropouts and a decrease in academic performance. Both the abovementioned health conditions are serious mental problems that university students are at heightened risk of experiencing. Although several prior studies confirmed positive relationships between burnout syndrome and social media overuse, so far little is known about the long-term relationship between these psychological characteristics of youths. In light of this, the purpose of this study was to examine the link between Facebook addiction and motives and changes in academic burnout indicators over time among university students.

### 1.1. Facebook—A Social Network Which Users Can Become Addicted to

The extensive rise of new technologies and electronic devices has dramatically changed our everyday life. Internet and social communication services assist in many daily activities, as they were created to facilitate better communication between people and help maintain relationships. However, in one study conducted by Błachnio and Przepiórka [3], entitled “Be Aware! If You Start Using Facebook Problematically You Will Feel Lonely: Phubbing, Loneliness, Self-esteem, and Facebook Intrusion. A Cross-Sectional Study”, the authors found that social media usage paradoxically may increase one’s feelings of loneliness and diminish satisfaction with life. Therefore, the dark side of online activity leads to the emergence of the concept of excessive social media addiction (social networking site (SNS) addiction). The availability of social network applications and the possibility of always being connected are some of the attraction factors that explain the reasons why some individuals use social networking sites excessively [4]. Rajesh and Rangaiah [5] stated that social media addiction is an overarching term that includes all social media applications, like Facebook, Twitter, Instagram, WhatsApp, YouTube, etc. However, the most popular of these is Facebook and scholars have primarily explored the antecedents and consequences of addiction to this network, also known as Facebook overuse, Facebook intrusion, Facebook use disorder, Facebook dependency, excessive use of Facebook, Facebook addiction disorder (FAD) and problematic use of Facebook [3,6,7,8,9]. Facebook addiction is defined as a form of Internet addiction and its core symptoms are excessive involvement in online activities and undisciplined use of Facebook, which result in problems with everyday functioning [7]. Additionally, based on Griffiths’ [10] Internet addiction symptoms, some authors also add to this definition characteristic features observed in diseases of addiction, e.g., tolerance (requiring increasing amounts of hours of media usage to achieve previous positive effects); withdrawal (unpleasant feelings when the media usage is discontinued or reduced); relapse (reverting to earlier patterns when trying to stop media usage or control usage); salience (preoccupation with and cravings for media usage that dominate thinking and behaviour); mood modification (worsening/improvement of or alteration in mood experiences when using media or when it is impossible to use it); and conflict (prioritising media usage over other actions, causing conflicts in relationships, work/education and all other activities) [5,6,10].

Analyses of Facebook users’ motives have resulted in several primary categories being distinguished by scholars, e.g., communication motives of FB use, such as social motives consisting of maintaining or establishing personal contacts with friends or family and building companionship; and non-communication uses of FB, such as instrumental motives (passing time, entertainment, playing games and information seeking), personal motives/importance (constructing an online self-identity and self-image) and escapism [11,12]. Non-communication motives for FB use are risk factors for Facebook addiction [13].

Being a heavy Facebook user disturbs not only the day-to-day offline functioning of a person, but also has a negative impact on somatic and mental health. Kuss and Griffiths [4] found that problematic social network site use may lead to a variety of adverse consequences, e.g., a decrease in real-life communication, worsening of academic performance, social conflicts and impoverishment of social bonds. One more distinguishing consequence and feature of people addicted to social media is lower self-esteem [6]. Many Facebook users try to meet the need to present one’s self in a more positive light in order to impress others and in this way boost self-esteem [5]. However, this triggers a mechanism where their positive self-esteem depends on their degree of popularity on social networking sites. Self-beliefs about being worse than others among FB over-users are often explained by a downward social comparison that causes negative feelings about the users themselves and a negative perception of their competence and attractiveness [14,15]. According to the literature, self-esteem is related to Facebook dependency because people compensate for their difficulties in real life social relations when using social media [8]. Social networking site (SNS) addictions may also lead to health problems (e.g., poor sleep quality), mental disorders (e.g., anxiety, depression, body image disorders, eating disorders, drinking disorders) or psychological imbalances as they strongly engage users’ time and energy, which may interfere with their real life duties and needs, as well as being associated with unhealthy behaviours and lower educational and personal achievements [11,16,17,18,19,20,21,22,23]. Specifically, Facebook addiction disorder (FAD) negatively impacts study habits and academic achievement, resulting in lower grades [24], and is related to higher academic anxiety [25]. However, studies in this area are scarce and show mixed results; González et al. [26] stated that FB usage has several positive outcomes in academic performance (e.g., FB as an effective teaching tool or a method that increases learning motivation) as well as negative results (e.g., attention deficits, drawbacks and the time that Facebook takes away from purely academic tasks). A few studies have confirmed the relation of online addiction (IA and FAD) with higher school stress and burnout [18,27,28]. It is worth noting that university students are a population at high risk of social media addiction due to four reasons: (1) they have notably high Internet literacy; (2) their online activities are not supervised by adults, i.e., teachers or parents; (3) they have flexible schedules and more free and unlimited access to online activities; and (4) social media offers an easy way for young adults to fulfil developmental tasks, e.g., interact with different people, build up intimate relationships and form their identities [29]. The abovementioned negative consequences of Facebook addiction indicate that students who are unable to control their Facebook use may experience other problems associated with their everyday educational performance and decrease their chances of academic success. Therefore, in this study we were concerned with the associations between Facebook addiction and changes in academic burnout level.

### 1.2. Burnout and Social Networking Site Addictions

Burnout is an ailment associated with chronic distress and depletion of personal resources that increases the risk of serious social, mental and physical problems. 

The three core elements of this complex phenomenon are: (1) overwhelming exhaustion, (2) cynicism and detachment and (3) a sense of personal inefficiency [30]. Although some authors suggest a high level stability for all three dimensions of burnout [31], the experience of burnout may change over time [30]. Thus, most recent theories of burnout focus on the process of change in this syndrome and the notion of chronic imbalances leading to strain. According to longitudinal studies there are several possible sequences for the development of this syndrome and burnout is related to future mental and physical problems [32]. According to the BAT model proposed recently by Schaufeli et al. [33], in states of burnout people feel extreme tiredness and suffer because of impairments to their emotional and cognitive regulation processes, which cause mental distancing as a self-protection reaction. Experiencing the abovementioned symptoms indicates a loss of control in major self-regulating systems and leads to secondary symptoms, e.g., psychological distress, psychosomatic complaints and depressed mood [34]. Indeed, numerous studies have confirmed that burnout syndrome at work or in education is associated with an increased likelihood of depression and anxiety disorders [35,36,37]. According to a meta-analysis by Salvagioni et al. [38], burnout is a predictor of 12 somatic diseases. Thus, burnout can have a multi-faceted, highly individualised psychological, social and somatic adverse impact on health. In addition, there is a link between burnout and other areas of mental health, such as eating disorders [39,40], alcohol and drug abuse [41], Internet addiction [28] and social media addiction [42,43]. A meta-analysis conducted by Madigan and Curran [44] confirmed that burnout leads to poorer school performance and worse academic achievement in school, college and university (reduced efficacy had the largest negative correlation with academic achievement).

Demerouti et al. [45], using the most popular burnout model JD-R, which has also been successfully applied in an educational context by Salmela-Aro and Upadayaya [46], stated that social media may alleviate job/school-related pressure and stress. From this perspective, social media may be one of the social recourses that reduces all negative stressors and emotional exhaustion, as well as enhancing the success of individuals thanks to support from online groups or co-workers (classmates) [47,48]. However, social media overuse has adverse effects and has been found to positively correlate with job and student burnout [18,27,48]. According to Walburg et al. [18], the link between burnout and Facebook addiction results from the escapist mechanism underlying social media use. This defence mechanism allows an individual to escape from and temporarily forget about problems that they have at school or work. However, like in any addiction, the dose of the agent, i.e., the length of Facebook use, must be constantly increased to achieve the desired effect—psychological balance. At the same time, due to the concentration of life around Facebook, the problems that a person is struggling with accumulate and become even more difficult to solve. Social media and Internet dependencies, according to some scholars, are examples of a process of self-undermining behaviour that causes maladjustment in the requirement cycle of job-/school-related demands. This process refers to all maladaptive coping behaviours that increase difficulties in meeting job/educational requirements, magnify conflicts and lead to more serious negative and unhealthy behaviours over time [49,50]. Another identified mechanism for social media overuse and job burnout is “social comparison”. Facebook users compare themselves to others and this process plays a mediating role, as those who are addicted make downward comparisons [48]. However, in such comparisons “the rich get richer and the poor get poorer”, meaning that Facebook users with low self-esteem will, due to these comparisons, see themselves as even more imperfect than others and more worthless [8]. As a consequence, they constantly feel negative emotions such as envy and social media use anxiety [42]. Negative emotions (e.g., anger, anxiety, irritation) experienced by people under stress narrow their thought action repertoires, indicating maladaptive regulation, as has been recently proposed in the JD-R model by Bakker and de Vries [50]. Specifically, the addiction demonstrates an inflexibility of behaviour that inhibits a person’s ability to adjust to a particular situation because of the same constantly repeated ineffective actions, e.g., drug taking, alcohol use, wasting time online, etc. According to Sriwilai and Charoensukmongkol [51], people who are addicted to social media tend to have lower mindfulness and more frequently use ineffective emotion-focused coping strategies. Reduction of the use of problem-oriented coping strategies triggers emotional exhaustion. Similarly, in the JD-R model, coping inflexibility refers to the inability (or reduced ability) to select an effective coping strategy, i.e., behaviour that correctly matches the situational demands and can monitor its own effectiveness. The vicious cycle of job/school demands presents overwhelming requirements that aggravate job- or school-related stress and problems and lead to burnout symptoms and other psychological and behavioural problems [50].

### 1.3. The Present Study

On the basis of the theory and empirical studies described earlier, two hypotheses are addressed in this longitudinal study. The first hypothesis deals with the prospective effects of changes in academic burnout and its dimensions and FB use motives and importance on time2 FB intrusion. More specifically, we proposed that changes in academic burnout dimensions (increases in exhaustion, cynicism and personal inefficacy; from time 1 to time 2), as well as high FB motives and importance (at times 1 and 2), would predict a higher time2 FB intrusion level (Hypothesis 1). We also hypothesised that including FB motives and importance in the regression model would enhance the prediction power of changes in academic burnout with regard to time 2 FB intrusion (Hypothesis 2a). Furthermore, in line with past studies, we hypothesised that the interactions between changes in academic burnout and its dimensions and FB motives and importance would explain a significant proportion of the variance in FB intrusion. These interactions would indicate how changes in academic burnout over time change the strength of the associations between FB motives and importance and FB intrusion (Hypothesis 2b). 

## 2. Materials and Methods

### 2.1. Study Population and Data Collection

The survey was conducted twice with a 4 month time interval among 130 university students aged between 19 and 23 years old (M = 20.41, SD = 0.93; two-wave longitudinal study). However, due to a lack of data, the statistical analyses included 115 results. Data were obtained from young adults studying psychology (second year of study) and teaching studies (first and second years) and of these respondents the majority were young females (85.2%). All students were invited to participate in the study by anonymously filling out a set of questionnaires twice: at the beginning and at the end of the winter semester of the 2019/2020 academic year. This study obtained ethical approval from the Commission of the Ethics Committee of the Pedagogical University of Krakow (WP BS-642/P/2019/20).

### 2.2. Measures

The Facebook Intrusion Scale (FIS) created by Elphinston and Noller [7] is used to measure addiction to Facebook. The scale includes eight statements (e.g., “I often think of Facebook when I do not use it”) to which the tested person responds on a seven-point Likert scale, where 1 = strongly disagree and 7 = strongly agree; the higher the score, the higher the intensity of addiction to Facebook. The questionnaire has good psychometric properties, with a Cronbach’s α = 0.84. In this study, α ranged between 0.85_T1_ and 0.88_T2_.

The Facebook Motives Scale created by Błachnio et al. [8] measures the level of motivation to use Facebook. The scale includes 21 statements (e.g., “I want to express and present myself”) to which the respondent answers on a seven-point Likert scale (1 = strongly disagree, 7 = strongly agree). The scale enables the measurement of three characteristics of FB users: social motives of FB use (α ranged between 0.51_T2_ and 0.57_T1_); instrumental motives for FB use (α ranged between 0.60_T1_ and 0.66_T2_); and personal importance of FB (α ranged between 0.83_T1_ and 0.85_T2_).

The Academic Burnout Scale (MBI-SS) created by Schaufeli et al. [52] measures students’ academic burnout. The scale includes 15 statements (e.g., “My studies exhaust me emotionally”) to which the respondent answers on a seven-point Likert scale, from 0 = never to 6 = every day. High scores on the Exhaustion (E) and Cynicism (C) subscales and low scores on the Academic Efficacy scale are interpreted as indicating the burnout of the person (personal inefficiency (PE)). In this study, α for the total MBI score ranged between 0.68_T1_ and 0.81_T2_ and for its three core dimensions it ranged between 0.86_T1_ and 0.89_T2_ (E), 0.69_T1_ and 0.70_T2_ (C) and 0.75_T1_ and 0.72_T2_ (PE).

### 2.3. Statistical Analysis

SPSS 22 with the macro PROCESS 3.0 package by Hayes [53] was used to analyse the data. The hypotheses were examined in hierarchical regression analyses. Several blocks of regression models were created in order to check the significance of interactions between burnout dimensions measured at time 1 (beginning of the winter semester (T1)) and time 2 (the end of the winter semester (T2)). For all three Facebook use motives, we first tested the blocks in which we included independent variables, i.e., Facebook motives and burnout dimensions collected at T1 and Facebook intrusion as dependent variable at T1 and T2. Secondly, we examined the regression models for Facebook intrusion at time 2, which were explained by the burnout dimension from time 2 and Facebook motives at time 1. Subsequently, the two-way interactions between the academic burnout dimensions measured at times 1 and 2 and the Facebook motives measured at time 1 were considered in the models. Following Cohen et al.’s [54] recommendations, before we analysed the interaction effects, all scores for the independent variables were standardised. The bootstrap method for interaction effects was used in order to confirm the significance of the results (model 1).The cross-lagged design of our study was characterized by the measurement of eight variables at two points in time (two-wave longitudinal studies with a 4 month time interval), making it possible to estimate within-time associations (e.g., variable X at T1 predicting variable Y at T2) [55]. The cross-lagged effects included the possibility of predicting the degree to which a dependent variable (FB intrusion) was influenced by other variables (academic burnout indicators and FB motives and importance) that had been measured before. Thus, the generated results can be interpreted as predicting relative changes (i.e., relative increases or decreases) in the outcome variable [56]. Additionally, we also controlled for whether cross-lagged effects occurred in both directions and assessed the significance and relative strength of the relationships [57].

## 3. Results

### 3.1. Testing the Prediction Power of Changes in Academic Burnout Dimensions over Time and FB Motives and Importance (Times 1 and 2) on FB Intrusion Measured at Time 2

Firstly, we examined the significance of changes in academic burnout. The *t* statistic revealed a significant increase in three indicators of academic burnout: exhaustion (*t*_(111)_ = −3.53, *p* = 0.001), cynicism (*t*_(111)_ = −4.91, *p* < 0.0001) and academic burnout total score (*t*_(111)_ = −3.67, *p* < 0.0001). The result for personal inefficiency was insignificant (*t*_(111)_ = 1.21, *p* = 0.230).

Regarding FB intrusion, as the results in Table 1 show, time 1 FB intrusion (beta = 0.47, *p* < 0.0001) accounted for 22% of the variance in time 2 FB intrusion (F_(1, 113)_ = 32.52, *p* < 0.001; block 1).The results of the regression analysis indicated that when controlling the FB intrusion measured at time 1, changes in the academic burnout total score did not significantly predict time 2 FB intrusion (B = 0.03, *p* = 0.510). In Table 1, we present the regression models for the changes in the sub-dimensions of academic burnout (block 2), along with the additive independent variables, FB motives and importance (blocks 3 and 4). The regression model with three dimensions of academic burnout revealed that changes in cynicism significantly predicted time 2 FB intrusion and explained 6% of the variances in the dependent variable (B = 0.41, *p* = 0.002, adjusted R^2^ = 0.28; block 2).

The prediction power of FB motives and importance for time 2 FB intrusion was analysed separately for each independent variable, as all were strongly associated with each other. The level of time 2 FB intrusion was significantly predicted by FB motives and importance at time 1 (FB social motive _T1_: B = 0.20, *p* < 0.05, F_(1, 113)_ = 4.50, *p* = 0.036,adjusted R^2^ = 0.03; FB instrumental motive _T1_: B = 0.27, *p* < 0.01, F_(1, 113)_ = 9.09, *p* = 0.003, adjusted R^2^ = 0.07; FB importance _T1_: B = 0.39, *p* < 0.001, F_(1, 113)_ = 20.44, *p* < 0.001, adjusted R^2^ = 0.15) and at time 2 (FB social motive _T2_: B = 0.37, *p* < 0.001, F_(1, 113)_ = 18.27, *p* < 0.001, adjusted R^2^ = 0.13; FB importance _T2_: B = 0.75, *p* < 0.001, F_(1, 113)_ = 150.29, *p* < 0.001, adjusted R^2^ = 0.57). Surprisingly, the regression model for the time 2 FB instrumental motive was insignificant. Testing the additive independent variables (i.e., FB motives and importance measured at time 1) did not improve the percentage of explained variables (28% of the explained variance, F_(7, 104)_ = 7.15, *p* < 0.0001) (block 3, Table 1). 

With regard to block 4, except for controlled time 1 FB intrusion, only two independent variables significantly predicted time 2 FB intrusion: time 2 personal importance of FB (B = 0.74, *p* < 0.0001) and decrease in exhaustion (B = −0.15, *p* < 0.05); both explained 35% of the variance in the dependent variable. Hypothesis 2a was partially supported, i.e., time 2 personal importance of FB enhanced the prediction power of changes in exhaustion and decreased the influence of changes in cynicism to an insignificant level.

Following the suggestions of Ford et al. [58], we also examined the potential dynamics of reverse causation effect, i.e., changes in FB intrusion and FB motives and importance (times 1 and 2) as predictors of time 2 academic burnout level (reversed hypothesis model). In the additional testing of stability coefficients (i.e., variable X at T1 predicting variable X at T2), time 1 academic burnout level significantly predicted time 2 academic burnout (beta = 0.50, *p* < 0.0001; adjusted R^2^ = 0.24 (F_(1, 111)_ = 36.61, *p* < 0.001)). All regression coefficients of the independent variables (i.e., changes in FB intrusion, FB motives and importance) in the three tested models for the academic burnout total score measured at time 2 were insignificant, indicating that there was no evidence for any reverse causation (changes in FB intrusion in model 1: B = −0.10, *p* = 0.487; in model 2, the examined changes in FB intrusion with FB motives and importance measured at time 1 were: B = −0.08, *p* = 0.601; and in Model 3, the examined changes in FB intrusion with FB motives and importance measured at time 2 were: B = 0.001, *p* = 0.996). 

### 3.2. Testing the Interactive Effects of Changes in Academic Burnout and FB Motives and Importance on the Level of FB Intrusion Measured at Time 2

Regarding changes in the academic burnout level over time and FB motives and importance at time 1, the interaction effects were insignificant (blocks 1–3, Table 2). It is worth noting that entering changes in academic burnout into the regression model enhanced the associations between FB social motives and personal importance (time 1) and time 2 FB intrusion and diminished the relationship between time 2 FB instrumental motive and FB intrusion _T2_. Examination of the interaction between changes in academic burnout and time 2 FB motives and importance revealed a significant interaction effect only for changes in the academic burnout level and social motives of FB use (B = 0.30, *p* < 0.0001, F_(4, 107)_ = 18.82, *p* < 0.0001). The full model explained 39% of the variance in time 2 FB intrusion and the interaction significantly improved the prediction of time 2 FB intrusion (9% of explained variance, F*_change_* = 16.48, *p* < 0.0001) (block 4, Table 2). The bootstrap method, developed by Hayes [24], confirmed the significance of the interaction effect (F = 6.90, *p* = 0.009, 95%CI [0.18; 1.28]). The results indicate that the observed increase in the academic burnout level over time enhanced the association between social motives of FB use and FB intrusion, while the observed decrease in the academic burnout level over time lowered the strength of this relationship (see Figure 1). 

In contrast, the interaction effects in models including the time 2 instrumental motive and personal importance of FB did not significantly predict time 2 FB intrusion (blocks 5 and 6, Table 2; hypothesis 2 was partially supported. It is worth adding that entering changes in academic burnout into the regression model reduced the prediction power of the FB social motive and personal importance for FB intrusion (all measured at time 2). 

Regarding our second hypothesis, the analyses also included the answer to the question concerning the extent to which changes in academic burnout dimensions and FB motives and importance were associated with time 2 FB intrusion (see Table 3). In block 1 (Table 3), it can be seen that changes in cynicism (B = 0.34, *p* < 0.0001) and personal inefficacy (B = −0.19, *p* < 0.05) as well as the interaction effect between the time 1 social motive of FB use and changes in cynicism (B = 0.22, *p* < 0.05), were significant predictors of time 2 FB intrusion (14% of the explained variance). However, the Hayes analysis did not confirm the significance of the interaction effect (F = 3.39, *p* = 0.068, 95%CI [−2.50; 0.09]). In block 2 (Table 3), it can also be seen that none of the interactive effects significantly predicted time 2 FB intrusion, and only changes in cynicism (B = 0.23, *p* < 0.05) appeared as a significant predictor of time 2 intrusion (3% of the explained variance).

The model examining the interactive effects of academic burnout dimensions and time 1 personal importance revealed significant interaction effects between time 1 personal importance and changes in exhaustion (B = 0.21, *p* < 0.05) as well as between time 1 personal importance and changes in personal inefficacy (B = −0.25, *p* < 0.01). Furthermore, 15% of the variance in the time 2 FB intrusion was explained by personal importance (B = 0.22, *p* < 0.01), changes in cynicism (B = 0.21, *p* < 0.05) and the interactions between the independent variables mentioned above; adding the interaction effects significantly improved the prediction power of the regression model (10% of the variance, F*_change_* = 5.52, *p* = 0.001). Additional deployment of Hayes’ bootstrap method confirmed the significance of both interaction effects, i.e., time 1 personal importance and changes in exhaustion (F = 6.38, *p* = 0.013, 95%CI [−0.07; 0.57]) and changes in personal inefficacy (F = 7.80, *p* = 0.006, 95%CI [−0.46; −0.08]). The results indicate that the observed increase in exhaustion and decrease in personal inefficacy levels over time enhanced the association between FB personal importance and FB intrusion, and the observed decrease in exhaustion and increase in personal inefficacy levels over time diminished this relationship (see Figure 2 and Figure 3).

In blocks 4–6 (Table 3), we present the results for the tests of the prediction power of the effects of the interactions between changes in academic burnout dimensions and time 2 FB motives and importance. The statistics revealed a significant interaction effect between time 2 social motives of FB use and changes in exhaustion (B = 0.25, *p* < 0.01). In the regression model, time 2 social motives of FB use (B = 0.22, *p* < 0.01) and cynicism (B = 0.21, *p* < 0.05), as well as the interaction effects mentioned above, significantly predicted time 2 FB intrusion (15% of the explained variance), and the interaction effects accounted for significant variance in the time 2 FB intrusion (10% of the variance, F*_change_* = 6.20, *p* = 0.001) (block 4, Table 3). However, the bootstrap method did not confirm the significance of the tested interaction effect (time 2 social motives of FB use x changes in exhaustion: F = 2.81, *p* = 0.097, 95%CI [−0.21; 0.88]). Block 5 (Table 3) shows that changes in cynicism appeared to be a significant predictor of time 2 FB intrusion (B = 0.23, *p* < 0.05) and highlights the interaction effect between the instrumental motive of FB use and changes in exhaustion (B = 0.22, *p* < 0.05). However, the Hayes analysis did not confirm the significance of this (time 2 instrumental motives of FB use × changes in exhaustion: F = 2.74, *p* = 0.101, 95%CI [−2.08; 0.19]). The model for personal importance of FB indicated non-significant interaction effects. Block 6 (Table 3) shows that only time 2 personal importance significantly predicted time 2 FB intrusion (B = 0.55, *p* < 0.0001, adj. R^2^ = 0.56). Hypothesis 2b was only supported for ∆ exhaustion, ∆ personal inefficacy and ∆ academic burnout total score (see Table 3). 

## 4. Discussion

Hypothesis 1 was only partially supported, as only changes in cynicism significantly predicted time 2 FB motives, and the time 2 instrumental motives of FB use appeared to be an insignificant predictor of time 2 FB intrusion. Our findings are only partially consistent with those of another study that suggested a positive relationship between Facebook addiction and student burnout [27]. In the study by Salmela-Aro et al. [27], which was a cross-lagged longitudinal study conducted among adolescents aged 12–18 years old, school burnout measured at time 1 predicted excessive Internet use at time 2, but only cynicism allowed for the prediction of later problematic Internet use. Moreover, the authors stated that youths in the 12–13 years old age group who scored high in cynicism could develop Internet and social media addiction by seeking meaning and value outside of school. Thus, according to the authors, the most effective way of supporting students’ mental health and preventing digital addictions is to enhance school engagement and intrinsic learning motivation and prevent school burnout. 

Secondly, we hypothesised that including FB motives and importance in the regression model would enhance the prediction power of changes in academic burnout over time for time 2 FB intrusion. This hypothesis was also partially supported, as the time 2 personal importance of FB enhanced the prediction power of changes in exhaustion and unexpectedly decreased the effect of changes in cynicism on FB intrusion measured at time 2 (the non-significant regression coefficient). More specifically, adding FB motives and importance to regression models showed that decreases in exhaustion significantly predicted a higher level of FB intrusion. Academic burnout emerges as an effect of difficulties in dealing with academic demands and the depletion of personal and social resources [49]. In other words, it evolves as a result of the collapse of regulatory systems that should protect the individual from the negative consequences of educational failure and strengthen students’ ability to cope with difficulties in academic settings. In terms of Jacobs’ [59] theory of addiction, the core reason for this development is because of distress that is not confronted and dealt with by individuals. According to this approach, a lack of effort in coping with everyday problems is due to low general self-efficacy. The only way to reduce negative states and distress is by increasingly compulsive Facebook activity, which, however, generates additional stress and, as a consequence, negatively affects the psychosocial functioning of the addicted person [60]. Taking into account our results, the use of Facebook for non-communication reasons, i.e., the personal importance of Facebook, seems to cause individuals to advance to the next stages of burnout. This is why individuals are not so cynical but more exhausted and more prone to negatively judge themselves as workers/students. 

The final study hypothesis assumed that the interactions between changes in academic burnout indicators and FB motives and importance would explain a significant proportion of the variance in the time 2 FB intrusion and change the strength of the associations between FB motives and importance and time 2 FB intrusion. Our findings also only partially confirmed this hypothesis. The effect suggested above was only confirmed for changes in the academic burnout total score, on the association between time 2 FB social motives and time 2 FB intrusion, and for changes in exhaustion and personal inefficacy, on the association between the personal importance of FB measured at times 1 and 2 and FB intrusion. Unexpectedly, an increase in personal inefficacy reduced the abovementioned relationship.

The positive effects of greater academic burnout and exhaustion seem to be consistent with results from past studies [27,48] and with the mechanisms connected to escapism from real life problems and reduced psychological distress thanks to online activity. In the JD-R model, burnout causes health impairment and lack of life satisfaction because of chronic dysregulation of core emotional and cognitive mechanisms [33,49]. Similarly, Facebook addiction leads to higher perceived stress and, as a long-term consequence, impoverished well-being and impaired general health [57]. Both conditions seem to reinforce each other and accelerate the process of loss of mental and physical balance. In this study, personal inefficacy had the opposite effect. The more the students felt incompetent in educational settings, the less Facebook was treated as a place where they could express themselves and the less severe the addiction to this form of social media was. We believe that this process may be related to the social comparison process. In addition, Han et al. [48] found that WeChat usage, downward social comparison and the interaction between them had significant positive effects on job burnout. According to Schaulfeli and Taris [61], professional inefficacy should be considered not as a sub-dimension of burnout but as an outcome of this process. It manifests the inability of users to positively evaluate themselves in work/education roles, e.g., low work or academic self-efficacy, feelings of incompetence in work or the field of study and a lack of productivity [62]. Meanwhile, Facebook addiction is related to higher social anxiety and lower general self-efficacy [56]. A study by Vivian [63] found that FB was used by students for informal learning via the use of status updates, private messaging, instant chat, tagging and FB groups. If the student feels a lack of professional efficacy in the studied field, this may heighten the pressure and stress connected to educational failure, and for this reason online activity may decrease. Our findings are also partially supported by the findings of Kalpidou et al. [64], as in their studies an increase in the number of friends on Facebook in the first years of university negatively affected academic and emotional harmony and was related to maladaptation at university. Problems with adaptation and coping with study-related stress may negatively impact the academic efficacy of students. It should also be taken into account that we conducted our study on social sciences students (from the psychology and teaching fields) who were mostly female. Students of these disciplines, especially females, are usually characterised by specific personality characteristics, e.g., greater openness to social contacts, a greater need for communication, more empathy, a willingness to share their own problems with others and seek help, etc. [65]. 

To sum up, our findings suggest that changes in academic burnout and FB motives and importance are suppressive variables, as including these variables in the regression model all together changed the significance of the relationship between the independent variables and FB intrusion. Moreover, when examining the relationship between FB motives and importance and FB intrusion, it is essential to consider changes in the academic burnout level, as this may be a key factor in understanding the mechanism underlying increasing social media addiction among students.

### Study Strengths and Limitations

Changes in burnout and its role in university student behaviours and outcomes are rarely analysed in longitudinal studies in the literature. Moreover, to the best of our knowledge, past studies have not tested Facebook intrusion, Facebook motives and importance and academic burnout indicators in longitudinal studies all together. Thus, our research sheds light on the mechanism underlying the associations between these psychological constructs and broadens the existing knowledge on these health-related conditions. The study design and longitudinal data provide information on the effects of academic burnout dynamics (e.g., relative increases or decreases in burnout and the process of energy depletion) connected to preparation for exams on the FB activity of students. However, our study also has some limitations. Firstly, the study sample was rather small and consisted of university students in the second year of study, most of whom were females from the teaching and psychology faculties. Future studies should examine the role of changes in academic burnout levels in Facebook addiction in all student groups, including different years of study as well different faculties. Gender differences should be also taken into account, as Facebook addiction, as well as burnout syndrome, have been found to be different depending on these sociodemographic characteristics. In addition, to increase the generalizability of the results, in future studies it would be useful to examine this relationship with a more cross-cultural study design. The data collected were obtained by means of an online survey with self-report questionnaires. However, the presence of a defence mechanism among addicted people causes underestimation of the symptoms and some negative consequences of Facebook use may be unnoticed or downplayed by the respondent. This is especially true since at the beginning, in the first stage of addiction, users can use Facebook intensively without feeling other symptoms of addiction [8]. Therefore, analyses based on qualitative data (e.g., interviews with a clinical sample) can provide in-depth information that helps in understanding the individual’s perspective (thoughts, emotions and behaviours), as well as more information about both concepts. In addition, a more objective perspective on the assessments of teachers, family and classmates could help in obtaining more detailed results. Finally, we based our results on the students’ declarations about online social media use and concentrated only on the Facebook app. However, during the study, participants reported that other applications enabling contact with others via the Internet are equally popular and that their use is even more common than Facebook. Therefore, it is possible that for some individuals we did not capture social media addiction symptoms because they were addicted to another application, and this may have interfered with our findings.

## 5. Conclusions

To our knowledge, the present study is among the first to examine the longitudinal associations between changes in academic burnout level, FB motives and importance and FB intrusion. The results are encouraging because the cross-lagged design of our study confirmed that FB intrusion measured at time 2 can be predicted by changes in academic burnout and its dimensions, which may interact with FB motives and importance. What is more, our study underlines the importance of investigating academic burnout among university students over time. The conclusions from this research may be used to propose preventive programs aimed at identifying students at risk of losing themselves in FB and burning out. A lack of relations in the real world can lead to escape into a virtual world. Many people treat online time as an easier form of ridding themselves of various difficult (for themselves) problems. Since the number of addicted Facebook users is rising and in the light of such addiction’s associations with burnout symptoms, there is a growing need for preventive interventions addressed to university students, e.g., organizing non-university events, internships or meetings; implementing programs such as check-in check-out or check & connect; or organizing support groups for students starting a new, more independent way of life (e.g., in a new city or among a new group of people).

## Figures and Tables

**Figure 1 ijerph-18-08055-f001:**
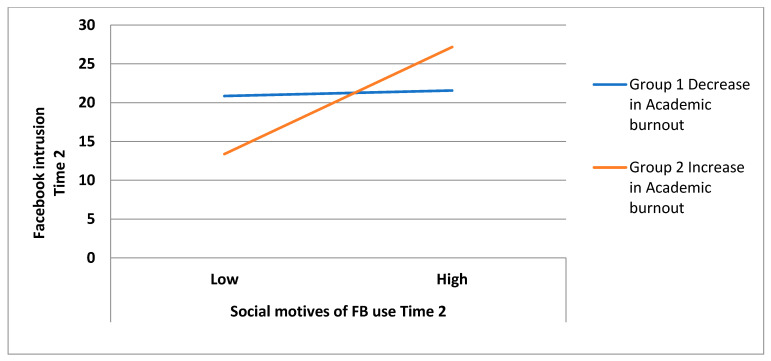
The effects of changes in academic burnout on the association between social motives for FB use and FB intrusion, both measured at time 2.

**Figure 2 ijerph-18-08055-f002:**
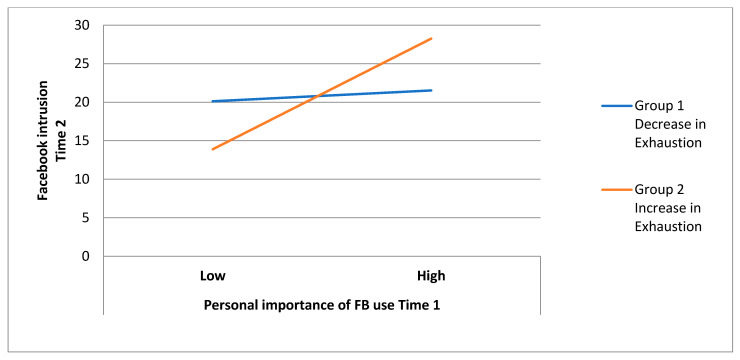
The effects of changes in exhaustion on the association between time 1 personal importance of FB use and time 2 FB intrusion.

**Figure 3 ijerph-18-08055-f003:**
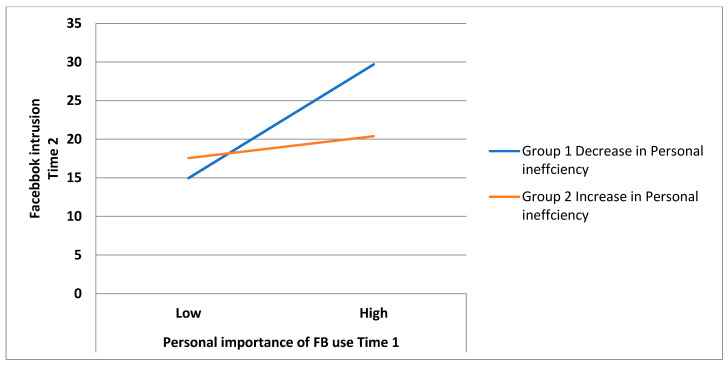
The effects of changes in personal inefficacy on the association between time 1 personal importance of FB use and time 2 FB intrusion.

**Table 1 ijerph-18-08055-t001:** Predictors of time 2 FB intrusion—multiple regression analysis results.

Predictors	Unstandardized Coefficient	SE	Standardized Coefficient	95%CI	
				*LL*	*UL*
Block 1					
FB intrusion _T1_	0.51	0.09	0.47 ***	0.33	0.68
F_(6, 105)_ = 320.52, *p* < 0.0001, R^2^ = 0.22 (adj. R^2^ = 0.22)					
Block 2					
FB intrusion _T1_	0.44	0.08	0.44 ***	0.28	0.61
∆ Exhaustion	−0.08	0.08	−0.09	−0.23	0.07
∆ Cynicism	0.41	0.13	0.29 **	0.16	0.66
∆ Personal inefficiency	−0.18	0.10	−0.14	−0.40	0.03
F_(4, 107)_ = 110.54, *p* < 0.0001, R^2^ = 0.30 (adj. R^2^ = 0.28)					
Block 3					
FB intrusion _T1_	0.35	0.13	0.35 **	0.09	0.61
Social motives of FB use _T1_	0.21	0.34	0.06	−0.46	0.87
Instrumental motives of FB use _T1_	0.44	0.28	0.14	−0.12	10.00
Personal importance of FB _T1_	0.06	0.16	0.05	−0.25	0.37
∆ Exhaustion	−0.06	0.08	−0.07	−0.22	0.09
∆ Cynicism	0.36	0.13	0.25 **	0.10	0.62
∆ Personal inefficiency	−0.18	0.11	−0.14	−0.41	0.04
F_(7, 104)_ = 70.15, *p* < 0.0001, R^2^ = 0.33 (adj. R^2^ = 0.28)					
Block 4					
FB intrusion _T1_	0.21	0.07	0.21 **	0.07	0.35
Social motives of FB use _T2_	0.38	0.25	0.11	−0.11	0.87
Instrumental motives of FB use _T2_	−0.28	0.20	−0.10	−0.67	0.11
Personal importance of FB _T2_	0.74	0.10	0.61 ***	0.55	0.93
∆ Exhaustion	−0.13	0.06	−0.15 *	−0.25	−0.01
∆ Cynicism	0.13	0.11	0.09	−0.08	0.35
∆ Personal inefficiency	−0.01	0.09	−0.01	−0.18	0.16
F_(7, 104)_ = 210.89, *p* < 0.0001, R^2^ = 0.60 (adj.R^2^ = 0.57)					

∆ Exhaustion, ∆ cynicism, ∆ personal inefficiency: changes in academic burnout dimension scores. * *p* < 0.05; ** *p* < 0.01; *** *p* < 0.001.

**Table 2 ijerph-18-08055-t002:** Results of the hierarchical regression analyses examining the effects of changes in academic burnout and FB motives and importance measured at times 1 and 2 on FB intrusion measured at time 2.

Variables	Time 2 FB Intrusion
Block 1 F, adj. R^2^ (R^2^ change)	F_(4, 107)_ = 8.02 ***; adj. R^2^ = 0.20 (0.00)
FB intrusion _T1_	0.44 ***
Social motives of FB use _T1_	0.07
∆ Academic burnout	0.05
Social motives of FB use _T1_ × ∆ academic burnout	0.02
Block 2 F, adj. R^2^ (R^2^ change)	F_(4, 107)_ = 100.51 ***; adj. R^2^ = 0.26 (0.02)
FB intrusion _T1_	0.45 ***
Instrumental motives of FB use _T1_	0.14
∆ Academic burnout	0.02
Instrumental motives of FB use _T1_ × ∆ academic burnout	0.15
Block 3 F, adj. R^2^ (R^2^ change)	F_(4, 107)_ = 80.11 ***; adj. R^2^ = 0.20 (0.001)
FB intrusion _T1_	0.39 ***
Personal importance of FB _T1_	0.11
∆ Academic burnout	0.06
Personal importance of FB _T1_ × ∆ academic burnout	0.03
Block 4 F, adj. R^2^ (R^2^ change)	F_(4, 107)_ = 180.82 ***; adj. R^2^ = 0.39 (0.09 ***)
FB intrusion _T1_	0.44 ***
Social motives of FB use _T2_	0.26 ***
∆ Academic burnout	0.05
Social motives of FB use _T2_ × ∆ academic burnout	0.30 ***
Block 5 F, adj. R^2^ (R^2^ change)	F_(4, 107)_ = 90.06 ***; adj. R^2^ = −0.23 (0.03)
FB intrusion _T1_	0.50 ***
Instrumental motives of FB use _T2_	0.05
∆ Academic burnout	0.03
Instrumental motives of FB use _T2_ × ∆ academic burnout	0.16
Block 6 F, adj. R^2^ (R^2^ change)	F_(4, 107)_ = 370.04 ***; adj. R^2^ = 0.57 (0.01)
FB intrusion _T1_	0.22 **
Personal importance of FB _T2_	0.60 ***
∆ Academic burnout	0.05
Personal importance of FB _T2_ × ∆ academic burnout	0.11

Note: the standardized regression coefficients (betas) are reported; ∆ academic burnout: changes in academic burnout score. ** *p* < 0.01; *** *p* < 0.001.

**Table 3 ijerph-18-08055-t003:** Results of hierarchical regression analyses examining the effects of changes in academic burnout dimensions and FB motives and importance measured at times 1 and 2 on the FB intrusion measured at time 2.

Variables	Time 2 FB Intrusion
Block 1 F, adj. R^2^ (R^2^ change)	F_(8, 103)_ = 70.09 ***; adj. R^2^ = 0.31 (0.05)
FB intrusion _T1_	0.42 ***
Social motives of FB use _T1_	0.09
∆ Exhaustion	−0.08
∆ Cynicism	0.34 ***
∆ Personal inefficiency	−0.19 *
Social motives of FB use _T1_ × ∆ exhaustion	−0.09
Social motives of FB use _T1_ × ∆ cynicism	0.22 *
Social motives of FB use _T1_ × ∆ personal inefficiency	0.02
Block 2 F, adj. R^2^ (R^2^ change)	F_(8, 103)_ = 70.35 ***; adj. R^2^ = 0.31 (0.04)
FB intrusion _T1_	0.42 ***
Instrumental motives of FB use _T1_	0.07
∆ Exhaustion	−0.17
∆ Cynicism	0.23 *
∆ Personal inefficiency	−0.10
Instrumental motives of FB use _T1_ × ∆ exhaustion	0.19
Instrumental motives of FB use _T1_ × ∆ cynicism	0.08
Instrumental motives of FB use _T1_ × ∆ personal inefficiency	−0.05
Block 3 F, adj. R^2^ (R^2^ change)	F_(8, 103)_ = 80.66 ***; adj. R^2^ = 0.36 (0.10 **); (F = 50.52, *p* = 0.001)
FB intrusion _T1_	0.39 **
Personal importance of FB _T1_	0.24 *
∆ Exhaustion	−0.01
∆ Cynicism	0.20 *
∆ Personal inefficiency	−0.16 *
Personal importance of FB _T1_ × ∆ exhaustion	0.23 **
Personal importance of FB _T1_ × ∆ cynicism	−0.04
Personal importance of FB _T1_ × ∆ personal inefficiency	−0.25 **
Block 4 F, adj. R^2^ (R^2^ change)	F_(8, 103)_ = 110.29 ***; adj. R^2^ = 0.43 (0.10 **); F = 60.20, *p* = 0.001
FB intrusion _T1_	0.42 ***
Social motives of FB use _T2_	0.22 **
∆ Exhaustion	−0.13
∆ Cynicism	0.21 *
∆ Personal inefficiency	−0.01
Social motives of FB use _T2_ × ∆ exhaustion	0.25 **
Social motives of FB use _T2_ × ∆ cynicism	0.14
Social motives of FB use _T2_ × ∆ personal inefficiency	−0.01
Block 5 F, adj. R^2^ (R^2^ change)	F_(8, 103)_ = 70.20 ***; adj. R^2^ = 0.31 (0.06 *); F = 30.01, *p* = 0.034
FB intrusion _T1_	0.47 ***
Instrumental motives of FB use _T2_	−0.08
∆ Exhaustion	−0.13
∆ Cynicism	0.23 *
∆ Personal inefficiency	−0.14
Instrumental motives of FB use _T2_ × ∆ exhaustion	0.22 *
Instrumental motives of FB use _T2_ × ∆ cynicism	−0.08
Instrumental motives of FB use _T2_ × ∆ personal inefficiency	0.07
Block 6 F, adj. R^2^ (R^2^ change)	F_(8, 103)_ = 180.64 ***; adj. R^2^ = 0.56 (0.01)
FB intrusion _T1_	0.21 **
Personal importance of FB _T2_	0.55 ***
∆ Exhaustion	−0.13
∆ Cynicism	0.07
∆ Personal inefficiency	0.03
Personal importance of FB _T2_ × ∆ exhaustion	−0.02
Personal importance of FB _T2_ × ∆ cynicism	0.15
Personal importance of FB _T2_ × ∆ personal inefficiency	−0.06

Note: the standardized regression coefficients (betas) are reported; ∆ exhaustion, ∆ cynicism, ∆ personal inefficiency: changes in academic burnout dimension scores. * *p* < 0.05; ** *p* < 0.01; *** *p* < 0.001.

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
