# Peer review of "Be Aware of Burnout! The Role of Changes in Academic Burnout in Problematic Facebook Usage among University Students"

_ijerph, 2021, doi:10.3390/ijerph18158055_

Round 1

Reviewer 1 Report

Thematically interesting study. It approaches burnout in a non-professional target audience - student population - in articulation with current issues related to the use of the internet and, in particular, social networks such as Facebook.
Assuming at the research level the intention of a longitudinal understanding of burnout, the confrontation of perspectives - state vs. process - should have been problematized explicitly at the theoretical level. An assessment tool consistent with the evolutionary vision of the phenomenon should also have been chosen. Could the MBI version used give a processual view (about eventual changes) of the burnout?
- Some recent publications on Internet addition, even published in the IJERPH journal, should be integrated.
- Small and very restrict and homogenous sample.
- “In this study α ranged between .68 and .89” - α scores under .70 should be rethought.
- At the empirical level, it manages to innovate by introducing contributions to a longitudinal analysis of the phenomena - two-wave longitudinal study with a 4 month time-interval.
- Authors said: "The study design and longitudinal data may provide some information on causal effects" - We cannot agree with this statement. The research done does not provide a causal analysis.
- Conclusion section should be rewritten and focalized in what could be concluded from the research major findings, instead of making some general and speculative remarks.
In general, a research with interest and merit for publication, after some improvements.

Author Response

Thematicallyinterestingstudy. It approachesburnout in a non-professional target audience - student population - in articulation with currentissuesrelated to the use of the internet and, in particular, social networks such as Facebook.
Assumingat the researchlevel the intention of a longitudinalunderstanding of burnout, the confrontation of perspectives - state vs. process – shouldhavebeenproblematizedexplicitlyat the theoreticallevel. Anassessmenttoolconsistent with the evolutionaryvision of the phenomenonshouldalsohavebeenchosen. Could the MBI version usedgive a processualview (abouteventualchanges) of the burnout?

MBI scale has been considered the standard tool for research in this field, and has been translated and validated in many languages (Maslach et al.,. 2016).  According to studies several possible sequences for the development of the three burnout dimensions (burnout defined as a process) as measured with the MBI have been suggested by several authors and most of these models used the original MBI measure of burnout (for review see Toppinen-Tanner book “Process of burnout: structure, antecedents, and consequences”. What is more, similarly to our methodology,  some of  prior investigations used this tool for measure changes in burnout e,g, Schanfelt et al., 2015; Pisant et al., 2016). In our opinion MBI accurately capture changes in burnout dimensions level. We add some information about burnout as a process. We would like to addthat the mainaim of the study was not the examination of state vs process but the changes in the level of academic burnout and its relationship to FB activity over time. Thus we do not paid a lot of attention to the evolutionary vision of the burnout phenomenon.

- Some recent publications on Internet addition, even published in the IJERPH journal, should be integrated.

Thank you for your suggestion. We included in the manuscript a reference to texts about Internet Addiction from IJERPH (e.g. item 20/ 26 in the bibliography) and improved all references.
- Small and veryrestrict and homogenoussample.

We are aware of this study limitation and include it in the manuscripit as on eof the limitation od current study. However, In Poland, 90% of women study psychology and the same proportion is in the teacher field – considering this our study sample is consistent with the gender distribution in the population of psychology and teacher students.
- “In this study α ranged between .68 and .89” - α scoresunder .70 should be rethought.We disagree with this statement. According to the literacy due to the small number of test items (the MBIscale consists of only 15 items; Social motives of FB are consistent with 3 items; Instrumental motives of FB also are consistent with 3 items ), α values ​​in the range of .45 - .60 are acceptable (Bretz & McClary, 2014; Taber, 2018).
- At the empirical level, it manages to innovate by introducing contributions to a longitudinal analysis of the phenomena - two-wavelongitudinalstudy with a 4 month time-interval.

We added information in the statistic analysis section aout cross – lagged design of the study

- Authors said: "The study design and longitudinal data may provide some information on causal effects" - We cannot agree with this statement. with this statement – the longitudinal study design allow to permit a causal effects; The research done does not provide a causal analysis.

We have changed this expression into: The study design and longitudinal data may provide some information on the effect of academic burnout dynamic (e.g. relative increase or decrease and the process of energy depletion connected to preparation to the exams on the FB activity of the students.

Additionally, we agree that longitudinal study designs per se are no 100% guarantee for drawing valid causal inferences, however despite longitudinal study designs drawbacks most researchers agree that they allow to have an insight into causal effect. According to Taris and Kompier (2003) The extent to which causal inferences can be made by conducting longitudinal studies depends on four conditions:

1.temporal ordering of the focal variables,

We agree that 4-month interval  may be consider as too short, however in our opinion considering students learning process during semester (we conducted our study at the beginning and at the end of the winter semester) it is interval that allows to have an insight in  the dynamic of the burnout level connected to day – to  - day academic duties and the process of energy depletion connected to preparation to the exams at the end of the semester

2.the strength of the statistical association between them,

We checked the associations between all variables

  1. theoretical plausibility of the presumed causal relationship,

We explained the suggested relationship by reference to BAT model and JD-R model of burnout

  1. exclusion of plausible rival hypotheses for this relationship

We tested the reverse hypothesis

Additionally, we do not claim that we tested causal effect, we only stated that our results give the insight to possible causal effect

- Conclusion sectionshould be rewritten and focalized in whatcould be concluded from the research major findings, instead of makingsomegeneral and speculativeremarks.
In general, a research with interest and merit for publication, aftersomeimprovements.

We have revised the conclusion section

Thank you for all comments and your time and work spend on our text, and we are open to make further changes if we have further suggestion

Reviewer 2 Report

This is a review of the manuscript titled "Be aware of burnout! The role of changes in academic burnout in problematic Facebook usage among university students". This study aims at investigating the impacts of academic burnout (exhaustion, cynicism, and personal inefficacy), motives for Facebook use (social motives and instrumental motives), and personal importance of Facebook use on Facebook intrusion using a two-wave longitudinal survey of university students. This manuscript could be improved if the following concerns are addressed:

1. The major problem is that the authors did not analyze the two-wave longitudinal data appropriately. Specifically, they used the change scores of exhaustion, cynicism, and personal inefficiency and T2 social motives for Facebook use, T2 instrumental motives for Facebook use, and T2 personal importance of Facebook use as predictors of T2 Facebook intrusion, without controlling for the effect of T1 Facebook intrusion (Table 1, 2, and 3). When analyzing the reverse causation effect, the authors used the changes scores of Facebook intrusion, motives for Facebook use, personal importance of Facebook use as predictors of T2 Facebook intrusion as predictors of T2 academic burnout, without controlling for T1 academic burnout (p. 5). I suggest the authors use the cross-lagged panel analysis (Selig & Little, 2012).

Reference

Selig, J. P., & Little, T. D. (2012). Autoregressive and cross-lagged panel analysis for longitudinal data. In B. Laursen, T. D. Little, & N. A. Card (Eds.), Handbook of developmental research methods (pp. 265–278). The Guilford Press.

2. The sample size mentioned in the Abstract (N = 115) and that mentioned in the Methods section (N = 130) were not consistent. Besides, which countries were the participants from?

3. The three dimensions of burnout (exhaustion, cynicism, and personal inefficacy) should be introduced in more detail in the Introduction section.

4. In the Abstract it is stated that, "Most previous research has paid more attention to the relationship between FB addiction and burnout level by conducting cross-sectional studies." (p. 1) Please describe those studies in the Introduction section.

5. Motives for Facebook use and personal importance of Facebook use were mentioned in the hypotheses. However, these two concepts were not described earlier. Please provide more literature review on these concepts.

6. The authors did not explain why they expect that academic burnout would interact with motives for Facebook use and personal importance of Facebook use to predict Facebook intrusion. Besides, it is stated that, "in line with past studies, the interactions between changes in academic burnout and its dimensions and FB motives and importance will explain a significant proportion of the variance in FB intrusion" (p. 3). Please provide more details about those past studies.

7. The authors did not mention how importance of Facebook use was measured. Moreover, the subscales (e.g., social motives, instrumental motives) of the Facebook Motives Scale was not described.

8. The specific reliability coefficients for all subscales at both time points should be reported.

9. The authors examined the changes in academic burnout and its three subscales (p. 4). However, the result regarding the change in personal inefficacy was not reported.

10. The authors conducted a number of separate moderated regression analyses. It is possible to include nine interaction terms (social motives × exhaustion, social motives × cynicism, social motives × personal inefficiency, instrumental motives × exhaustion, instrumental motives × cynicism, instrumental motives × personal inefficiency, importance of Facebook use × exhaustion, importance of Facebook use × cynicism, and importance of Facebook use × personal inefficiency) in a single regression model.

11. There are typos and grammatical errors throughout the manuscript. The manuscript needs to be edited carefully.

Author Response

Thisis a review of the manuscripttitled "Be aware of burnout! The role of changes in academicburnout in problematic Facebook usageamonguniversitystudents". Thisstudyaimsatinvestigating the impacts of academicburnout (exhaustion, cynicism, and personalinefficacy), motives for Facebook use (socialmotives and instrumentalmotives), and personalimportance of Facebook use on Facebook intrusionusing a two-wavelongitudinalsurvey of universitystudents. Thismanuscriptcould be improvedif the followingconcernsareaddressed:

  1. The major problem isthat the authorsdid not analyze the two-wavelongitudinal data appropriately. Specifically, theyused the changescores of exhaustion, cynicism, and personalinefficiency and T2 socialmotives for Facebook use, T2 instrumentalmotives for Facebook use, and T2 personalimportance of Facebook use as predictors of T2 Facebook intrusion, without controlling for the effect of T1 Facebook intrusion (Table 1, 2, and 3).
  2. When analyzing the reverse causation effect, the authors used the changes scores of Facebook intrusion, motives for Facebook use, personal importance of Facebook use as predictors of T2 Facebook intrusion as predictors of T2 academicburnout, without controlling for T1 academicburnout (p. 5). I suggest the authorsuse the cross-lagged panel analysis (Selig& Little, 2012).

Reference

Selig, J. P., & Little, T. D. (2012). Autoregressive and cross-lagged panel analysis for longitudinal data. In B. Laursen, T. D. Little, & N. A. Card (Eds.), Handbook of developmentalresearchmethods (pp. 265–278). The Guilford Press.

Thank you, we added suggested analysis.

  1. The samplesizementioned in the Abstract (N= 115) and thatmentioned in the Methodssection (N= 130) were not consistent. Besides, whichcountrieswere the participants from?

Thank for the suggestion we added the information that the statistical analysis were conducted on 115 results because of lack of data. We also added the information that the research was conducted in Poland, which was included in the information on the ethics committee.

  1. The threedimensions of burnout (exhaustion, cynicism, and personalinefficacy) should be introduced in moredetail in the Introductionsection.

In the introduction, information about the three dimensions of burnout was supplemented. thank you for this comment.

  1. In the Abstractitisstatedthat, "Most previousresearchhaspaidmoreattention to the relationshipbetween FB addiction and burnoutlevel by conducting cross-sectionalstudies." (p. 1) Pleasedescribethosestudies in the Introductionsection.

Thank you for paying attention, we have expanded the introduction to include this information.

  1. Motives for Facebook use and personalimportance of Facebook usewerementioned in the hypotheses. However, thesetwoconceptswere not describedearlier. Pleaseprovidemoreliteraturereview on theseconcepts.

Thank you for paying attention, we have expanded the introduction to include this information.

  1. The authorsdid not explainwhytheyexpectthatacademicburnoutwouldinteract with motives for Facebook use and personalimportance of Facebook use to predict Facebook intrusion. Besides, itisstatedthat, "in line with past studies, the interactionsbetweenchanges in academicburnout and itsdimensions and FB motives and importancewillexplain a significantproportion of the variance in FB intrusion" (p. 3). Pleaseprovidemoredetailsaboutthose past studies.

We improved the introduction section in order to justify our study hypothesis

  1. The authorsdid not mentionhowimportance of Facebook use was measured. Moreover, the subscales (e.g., socialmotives, instrumentalmotives) of the Facebook MotivesScale was not described.

The FB personal importance is a subscale of FB motives and importance by Przepiórka and BÅ‚achnio and was described by us in the section of measures.

  1. The specificreliabilitycoefficients for allsubscalesatbothtimepointsshould be reported.

specific reliability coefficients for all subscales were given.

  1. The authorsexamined the changes in academicburnout and itsthreesubscales (p. 4). However, the resultregarding the change in personalinefficacy was not reported.

We do not understand this comments. In table 1 and 3 we analyzed changes in all three dimensions (and the ∆Personal inefficiency was also reported in both of them). In table 2 we analyzed changes in academic burnout total score (∆Academic burnout) so the ∆Personal inefficiency were not included into analyzes.

  1. The authorsconducted a number of separatemoderatedregressionanalyses. It ispossible to includenineinteractionterms (socialmotives × exhaustion, socialmotives × cynicism, socialmotives × personalinefficiency, instrumentalmotives × exhaustion, instrumentalmotives × cynicism, instrumentalmotives × personalinefficiency, importance of Facebook use × exhaustion, importance of Facebook use × cynicism, and importance of Facebook use × personalinefficiency) in a single regression model.

First, the sample size is too small to conduct such analysis. Secondly, the variables are intercorrelated so we decided to computed analysis for each FB motives and importance separately. Third , we believe that testing separately each condition of FB motives we may derive more precise information about its association with the rest tested variables.

  1. There are typos and grammatical errors throughout the manuscript. The manuscriptneeds to be editedcarefully.

The text has been revised linguistically.

Thank you for all comments and your time and work spend on our text, and we are open to make further changes if we have further suggestion

Reviewer 3 Report

Dear authors,

Your manuscript is interesting but I need you to answer some questions:

INTRODUCTION

  • The introduction should not be to have subsections.
  • Authors should include some epidemiological data to support their claims.
  • The authors have not stated the objective of the investigation.

MATERIALS AND METHODS

Study Population and Data Collection:

  • Page 3, line 106: The authors say they have done an "A two-wave longitudinal study". I don't know what this is. The authors should explain it better.
  • What was the target population? How was the sample chosen? The authors must specify it.
  • Authors must state which ethics committee they have consulted and the reference assigned to them.

REFERENCES

  • Many bibliographies are obsolete. The bibliographic citations used are more than 5 years old (53 %). The authors must update and arrange the bibliography.
  • Some references are incomplete or have errors. The authors should review this section.

Author Response

Yourmanuscriptisinteresting but I needyou to answersomequestions:

 Thank you for the positive evaluation of our work.

INTRODUCTION

  • The introductionshould not be to havesubsections.

Due to the specificity of the variables, we decided to leave the division into subsections.

  • Authorsshouldincludesomeepidemiological data to supporttheirclaims.

We addend such information

  • The authorshave not stated the objective of the investigation.

We do not understand this comment – the objective of the study was included twice In the manuscript: “the purpose of this study was to examine the link between Facebook addiction and mo-tives and changes in academic burnout indicators over time among university students.” And in the section “1.3. The present study”

MATERIALS AND METHODS

StudyPopulation and Data Collection:

  • Page 3, line 106: The authorssaytheyhavedonean "A two-wavelongitudinalstudy". I don' knowwhatthisis. The authorsshouldexplainitbetter.

In the line 209 we write: “A two-wave longitudinal study with a 4 month time-interval was conducted among.” Longitudinal study is a study conducted after the time period determined by the researcher.

We also added the suggested information in the statistical analysis section.

  • What was the target population? How was the samplechosen? The authorsmustspecifyit.

Target population – psychology students. The sample of respondents was selected for those willing to take part in the study.

  • Authorsmuststatewhichethicscommitteetheyhaveconsulted and the referenceassigned to them.

 Thank you for your attention, we have corrected the lack of information.

REFERENCES

  • Many bibliographiesareobsolete. The bibliographiccitationsusedaremorethan 5 yearsold (53 %). The authorsmust update and arrange the bibliography.

Citations have been updated by us, more publications under 5 years old.

  • The authorsshouldreviewthissection.

Bibliography and notation have been unified.

Thank you for all comments and your time and work spend on our text, and we are open to make further changes if we have further suggestion

Round 2

Reviewer 1 Report

The authors addressed the majority of the suggestions made to the first draft and improved the presented paper. So, we recommend the publication of this new version of the document.

Author Response

Thank you for your recommendation and all work and time you spend on our manuscript revision.

Reviewer 2 Report

This is a review of the revised version of the manuscript titled "Be aware of burnout! The role of changes in academic burnout in problematic Facebook usage among university students". The manuscript has been improved but there are some remaining issues to be addressed.

1. The authors did not conduct the cross-lagged panel analysis appropriately. The autoregressive effect (the effect of the outcome at T1 on itself at T2) should be controlled for (Selig & Little, 2012).

Reference

Selig, J. P., & Little, T. D. (2012). Autoregressive and cross-lagged panel analysis for longitudinal data. In B. Laursen, T. D. Little, & N. A. Card (Eds.), Handbook of developmental research methods (pp. 265–278). The Guilford Press.

2. The authors examined the changes in academic burnout and its three subscales. Specifically, the authors report that, "Firstly, we examined the significance of changes in academic burnout. The t statistic revealed a significant increase in three indicators of academic burnout, e.g. exhaustion (t(111)= -3.53, p = .001), cynicism (t (111)= -4.91, p<.0001), and academic burnout total score (t(111) = -3.67, p < .0001)." However, the result regarding the change in personal inefficacy was not reported.

Author Response

  1. The authors did not conduct the cross-lagged panel analysis appropriately. The autoregressive effect (the effect of the outcome at T1 on itself at T2) should be controlled for (Selig & Little, 2012).

We corrected the analysis by controlling outcome at T1 for each regression model. Thank you very much for drawing our attention to these errors and we are open to suggestions in the event of further corrections.

  1. The authors examined the changes in academic burnout and its three subscales. Specifically, the authors report that, "Firstly, we examined the significance of changes in academic burnout. The t statistic revealed a significant increase in three indicators of academic burnout, e.g. exhaustion (t(111)= -3.53, p = .001), cynicism (t (111)= -4.91, p<.0001), and academic burnout total score (t(111) = -3.67, p < .0001)." However, the result regarding the change in personal inefficacy was not reported.

We added the results for personal inefficiency.

Reviewer 3 Report

Dear authors,

Thanks for your reply. The explanations of the authors are satisfactory. The paper has greatly improved its quality.

However, you must bear in mind the following: - I know what a "longitudinal study" is. What I don't know is what a "two-wave longitudinal study" is. You must explain it better in the methodology.

Congratulations on your work.

Best regards

Author Response

Thank you very much for your recommendation and kind reply.